# Point-of-care diagnostic tests for influenza in the emergency department: A cost-effectiveness analysis in a high-risk population from a Canadian perspective

Stephen Mac[1,2☯]*, Ryan O'Reilly[1,2,3☯], Neill K. J. Adhikari[1,4,5], Robert Fowler[1,4,5], Beate Sander[1,2,6,7]

1 Institute of Health Policy, Management and Evaluation (IHPME), University of Toronto, Toronto, Canada, 2 Toronto Health Economics and Technology Assessment (THETA) Collaborative, University Health Network, Toronto, Canada, 3 Department of Medicine, McMaster University, Hamilton, Canada, 4 Department of Critical Care Medicine, Sunnybrook Health Sciences Centre, Toronto, Canada, 5 Interdepartmental Division of Critical Care Medicine, University of Toronto, Toronto, Canada, 6 ICES, Toronto, Canada, 7 Public Health Ontario, Toronto, Canada

☯ These authors contributed equally to this work.
* sm.mac@mail.utoronto.ca

**Data Availability Statement:** All relevant data are within the manuscript and its Supporting Information files.

## Abstract

### Background

Our objective was to assess the cost-effectiveness of novel rapid diagnostic tests: rapid influenza diagnostic tests (RIDT), digital immunoassays (DIA), rapid nucleic acid amplification tests (NAAT), and other treatment algorithms for influenza in high-risk patients presenting to hospital with influenza-like illness (ILI).

### Methods

We developed a decision-analytic model to assess the cost-effectiveness of diagnostic test strategies (RIDT, DIA, NAAT, clinical judgement, batch polymerase chain reaction) preceding treatment; no diagnostic testing and treating everyone; and not treating anyone. We modeled high-risk 65-year old patients from a health payer perspective and accrued outcomes over a patient's lifetime. We reported health outcomes, quality-adjusted life years (QALYs), healthcare costs, and net health benefit (NHB) to measure cost-effectiveness per cohort of 100,000 patients.

### Results

Treating everyone with no prior testing was the most cost-effective strategy, at a cost-effectiveness threshold of $50,000/QALY, in over 85% of simulations. This strategy yielded the highest NHB of 15.0344 QALYs, but inappropriately treats all patients without influenza. Of the novel rapid diagnostics, NAAT resulted in the highest NHB (15.0277 QALYs), and the least number of deaths (1,571 per 100,000). Sensitivity analyses determined that results

**Funding:** This work was partially supported by the World Health Organization. The funders had no role in study design, data collection and analysis, decision to publish, or preparation of the manuscript. S. M. is supported by a CIHR Frederick Banting and Charles Best Canada Graduate Scholarship Doctoral Award GSD-159274. B.S. is supported by a Canada Research Chair in Economics of Infectious Diseases (CRC-950-232429). There was no additional external funding received for this study.

**Competing interests:** NKJA co-chaired the WHO Guideline Development Group – Clinical Management of Severe Influenza Infections. This does not alter our adherence to PLOS ONE policies on sharing data and materials. All remaining authors have no conflicts of interest to declare.

were most impacted by the pretest probability of ILI being influenza, diagnostic test sensitivity, and treatment effectiveness.

## Conclusions

Based on our model, treating high-risk patients presenting to hospital with influenza-like illness, without performing a novel rapid diagnostic test, resulted in the highest NHB and was most cost-effective. However, consideration of whether treatment is appropriate in the absence of diagnostic confirmation should be taken into account for decision-making by clinicians and policymakers.

## Introduction

The influenza virus causes epidemics of acute respiratory illness, resulting in significant morbidity and mortality every year. Globally, annual epidemics contribute to approximately 3 to 5 million individuals developing severe illnesses, and 290,000 to 650,000 respiratory-related deaths [1]. Young children, older adults and patients with chronic or immunocompromising conditions are the groups at highest risk of infection, hospitalization, and severe outcomes (e.g., requiring critical care, or death) [2].

Even in high-income nations such as Canada, annual influenza results in approximately 12,000 (39 per 100,000) hospitalizations and 3,500 (11.3 per 100,000) deaths [3]. In individuals age 65 and over, the hospitalization rate increases to a six-year (2013 to 2019) average of 143 per 100,000 [4]. In a Cochrane review, oseltamivir, the most commonly administered antiviral neuraminidase inhibitor (NAI), was shown to reduce the time to symptom alleviation by 16.8 hours in adults and 29 hours in children when promptly administered, emphasizing the importance of rapid and accurate diagnosis of influenza [5]. Clinical judgement to diagnose influenza can be difficult due to the non-specific symptoms relative to other acute respiratory infections [6], making rapid diagnostic tests a valuable option to appropriately diagnose, with a high degree of accuracy, and start antiviral treatment [7, 8].

Until recently, the only method to confirm an influenza infection was the use of reverse transcriptase polymerase chain reaction (RT-PCR) [9]. However, RT-PCR tests are typically run in batches, resulting in turnaround times that can extend up to 24 hours and longer, thereby preventing health-care providers from using the diagnostic information to guide initial treatment. Several rapid diagnostic tests have been developed: traditional rapid influenza diagnostic tests (RIDT), digital immunoassays (DIA), and rapid nucleic acid amplification tests (NAAT). Each of these tests is relatively simple to administer and provides results within 30 minutes. In a recent meta-analysis conducted by Merckx and colleagues, all three categories of tests were associated with high overall (adults and children) specificities (>98%) for influenza A and B, with NAATs having the highest sensitivity (92% influenza A, 95% influenza B), followed by DIAs (80% influenza A, 77% influenza B) and RIDTs (54% influenza A, 53% influenza B) for the combined adult and children population [10].

While rapid diagnostic tests offer quicker results that could potentially inform treatment decisions, the cost-effectiveness of these tests in clinical environments (e.g. hospitals) is uncertain. The objective of this study was to assess the cost-effectiveness of these testing strategies for high-risk patients presenting to the emergency department (ED) with influenza-like-illness (ILI) using decision-analytic modeling from a health payer perspective. Evidence generated from this study can support seasonal influenza management and guide decisions about

applying rapid testing for influenza in clinical practice. We did not consider the importance of distinguishing between influenza and COVID-19 or a future pandemic virus in this study.

## Methods

### Model structure

We developed a decision-analytic model to assess the cost-effectiveness of diagnostic testing strategies on the outcomes of 65 year-old patients presenting to the emergency department (ED) with symptoms or signs that could be suggestive of influenza, which we hereafter label as ILI (Fig 1). We modeled 65 year-old patients in the base case to approximate populations at high-risk for severe influenza illness, recognising that this is not the only high-risk population. We assessed five overall strategies: (1) no test, and do not treat patients ("Don't Treat Anyone"), (2) no test and treat everyone ("Treat Everyone"), (3) rapidly test all patients with ILI and treat with NAI, (4) Batch PCR test, and treat until results become available ("Batch PCR–Treat"), and (5) Batch PCR test, but do not treat until results are available ("Batch PCR–Wait"). For strategy (3), four diagnostic methods were evaluated: (A) "RIDT", (B) "DIA", (C) "NAAT", and (D) "Clinical Judgement". All modelling and analyses were conducted using TreeAge Pro 2019 (TreeAge Software, Inc., Williamstown, MA).

The model simulated a disease history and care pathway for patients presenting to the ED with ILI, altering the probability of hospitalization, ICU admission and death based on the timing and appropriateness of treatment (Fig 1). Patients present to the ED with ILI and have a prior probability of influenza or another acute respiratory infection based on the known community prevalence of influenza. Cases of influenza were further defined as influenza A or B based on surveillance data. Patients testing positive (true or false positive) were assumed to all receive a regimen of NAI treatment, while patients testing negative did not. It was assumed that patients with a false negative result did not receive NAI therapy at any point during their

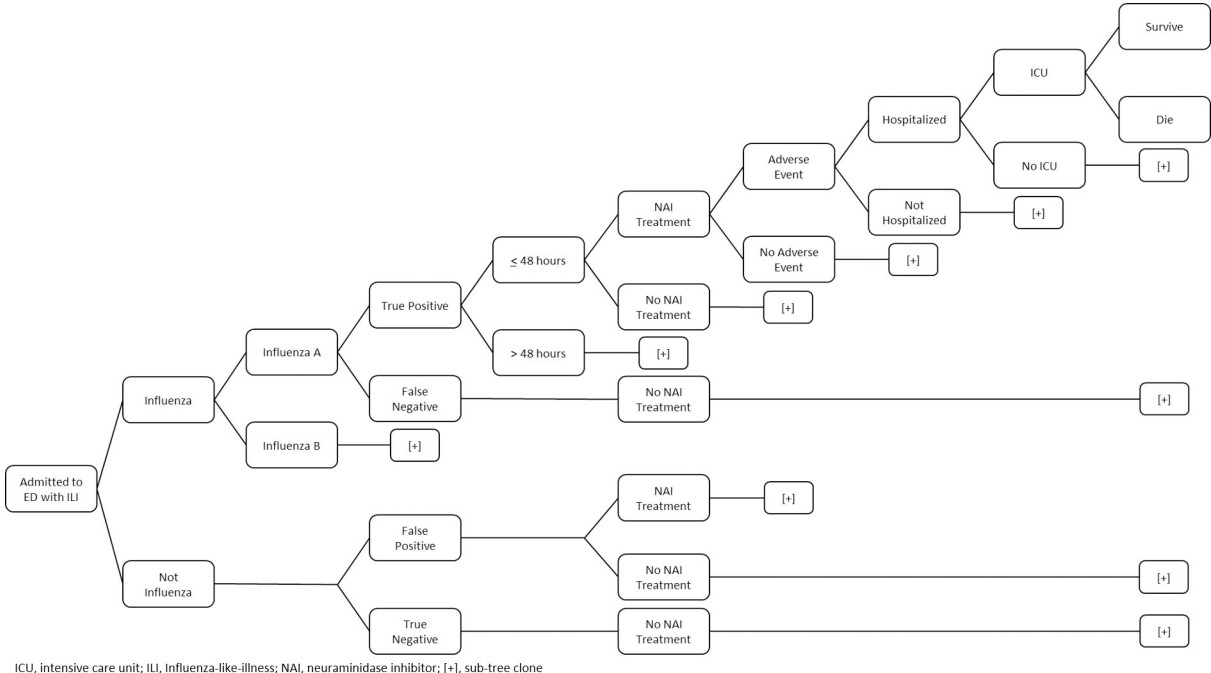

ICU, intensive care unit; ILI, Influenza-like-illness; NAI, neuraminidase inhibitor; [+], sub-tree clone

**Fig 1. Model schematic of decision-analytic model for high-risk patients presenting at ED.**

hospitalization. While some individuals testing negative may still receive NAI therapy, we assumed that they did not in the base-case. However, we examined this assumption in a scenario analysis where 50 percent of individuals testing negative for influenza still received NAI therapy. From the ED, patients could discharged home, or be admitted to the general ward or ICU, with cases admitted to the ICU having a higher probability of mortality. True positives treated with NAI were assigned a lower risk of mortality (see Table 1), as well as a decrease in the duration of symptoms based on data from recent meta-analyses [5, 8].

**Table 1. Key parameters for base-case.**

| Variable | Base-case value | Range | Source |
|---|---|---|---|
| **Diagnostic Tests** | | | |
| **Influenza A** | | | |
| Sensitivity. Adults | | | |
| RIDT | 0.426 | 0.348–0.509 | Merckx 2017 [10] |
| DIA | 0.754 | 0.666–0.826 | Merckx 2017 [10] |
| NAAT | 0.874 | 0.711–0.956 | Merckx 2017 [10] |
| Clinical Judgement | 0.36 | 0.22–0.52 | Dugas 2015 [16] |
| Batch PCR | 0.95 | 0.75–1[†] | Assumption (Merckx 2017) [10] |
| Specificity, Adults | | | |
| RIDT | 0.995 | 0.986–0.998 | Merckx 2017 [10] |
| DIA | 0.967 | 0.947–0.98 | Merckx 2017 [10] |
| NAAT | 0.98 | 0.932–0.995 | Merckx 2017 [10] |
| Clinical Judgement | 0.78 | 0.72–0.83 | Dugas 2015 [16] |
| Batch PCR | 0.95 | 0.75–1[†] | Assumption (Merckx 2017) [10] |
| **Influenza B** | | | |
| Sensitivity, Adults | | | |
| RIDT | 0.332 | 0.199–0.507 | Merckx 2017 [10] |
| DIA | 0.57 | 0.395–0.716 | Merckx 2017 [10] |
| NAAT | 0.757 | 0.518–0.907 | Merckx 2017 [10] |
| Clinical Judgement | 0.36 | 0.22–0.52 | Dugas 2015 [16] |
| Batch PCR | 0.95 | 0.75–1[†] | Assumption [10] |
| Specificity. Adults | | | |
| RIDT | 0.999 | 0.994–1 | Merckx 2017 [10] |
| DIA | 0.988 | 0.975–0.995 | Merckx 2017 [10] |
| NAAT | 0.993 | 0.978–0.998 | Merckx 2017 [10] |
| Clinical Judgement | 0.78 | 0.72–0.83 | Dugas 2015 [16] |
| Batch PCR | 0.95 | 0.75–1[†] | Assumption [10] |
| **ILI and Influenza-Related Probabilities** | | | |
| Pre-test probability of influenza | 0.144 | 0–1 | Seasonal assumptions [17] |
| Influenza A (Influenza B[‡]) | 0.873 | 0–1 | Seasonal assumptions [17] |
| Hospitalization | 0.116 | 0.09–0.15[†] | Ng 2018 [18] |
| ICU Hospitalization, ≥ 65y | 0.134 | 0.1–0.17[†] | CIRN (FluWatch) [19] |
| Tx within 48 hrs of symptom onset | 0.481 | 0.36–0.6[†] | Muthuri 2014 [8] |
| Adverse events, Tx | 0.075 | 0.056–0.094[†] | Santesso 2019 (Unpublished) |
| Adverse events, no Tx | 0.027 | 0.02–0.034[†] | Santesso 2019 (Unpublished) |
| Mortality (ICU admitted, Early Tx) | 0.276 | 0.21–0.35[†] | Muthuri 2014 [8] |
| Mortality (ICU-admitted, Late Tx) | 0.3198 | 0.24–0.4[†] | Muthuri 2014 [8] |
| Mortality (ICU-admitted, No Tx) | 0.5344 | 0.4–0.67[†] | Muthuri 2014 [8] |
| Mortality (Non-ICU, Early Tx) | 0.0809 | 0.06–0.1[†] | Muthuri 2014 [8] |

*(Continued)*

**Table 1.** (Continued)

| Variable | Base-case value | Range | Source |
|---|---|---|---|
| **Diagnostic Tests** | | | |
| Mortality (Non-ICU, Late Tx) | 0.1218 | 0.09–0.15[†] | Muthuri 2014 [8] |
| Mortality (Non-ICU, No Tx) | 0.1218 | 0.09–0.15[†] | Muthuri 2014 [8] |
| **Utilities (QALYs)** | | | |
| Population, age dependent | 0.88–0.94 | 0.8722–0.9426 | Mittmann 1999 [20] |
| QALYs lost for ILI (disutility), ≥ 65y | 0.0293 | 0.0233–0.0349 | Sander 2009 [21] |
| QALY improvement for symptom alleviation from treatment, > 18y | 0.00166 | 0.0012–0.0021[†] | Assumption (Jefferson 2015) [5] |
| Adverse event (disutility) | 0.0113 | 0.008–0.014[†] | Greiner 2006 [22] |
| **Costs** | | | |
| RIDT, per test | 20 | 20–26 | Merckx 2017 [10] |
| DIA, per test | 20 | 20–26 | Merckx 2017 [10] |
| NAAT, per test | 40 | 40–130 | Merckx 2017 [10] |
| Batch PCR, per test | 58 | 28–88 | Soto 2016 [23] |
| Emergency department visit | 468 | 351–585[†] | Ng 2018 [18] |
| Hospitalization | 7,977 | 5,983–9,971[†] | Ng 2018 [18] |
| ICU Hospitalization | 11,875 | 8,906–14,844[†] | Ng 2018 [18] |
| Oseltamivir treatment | 42 | 34–42 | Ontario Drug Benefit [24] |

[†] Uncertainty of key parameter was not reported and ± 25% was used to create a plausible range.

[‡] Probability of influenza B was complementary to probability of influenza A

CIRN, Canadian Immunization Research Network; DIA, digital immunoassay; ICU, intensive care unit; ILI, influenza-like-illness; NAAT, nucleic acid amplification test; PCR, polymerase chain reaction; QALY, quality-adjusted life year; RIDT, rapid influenza diagnostic test; Tx, treatment; y, years of age

Our model used a single healthcare payer perspective (applicable to each province in Canada) and lifetime time horizon to capture the potential benefits of averted mortality through optimal therapy for patients. Our model reported health outcomes (proportion of patients treated appropriately, adverse events, and mortality), quality-adjusted life years (QALYs), total healthcare costs, and net health benefit (NHB) to measure cost-effectiveness. Costs and QALYs were discounted at an annual rate of 1.5% as recommended [11].

## Net health benefit

In cost-effectiveness analysis, there are various units of measure used to present cost-effectiveness. Common outcomes include incremental cost-effectiveness ratios (ICERs) that take a cost-utility approach and express value in a $/QALY gained, net monetary benefit (NMB) that expresses value in terms of costs, and net health benefits (NHB) that expresses value in terms of the health outcome chosen (i.e., QALYs in this study).

As the number of strategies being compared increases, the ratio statistics of the ICER become more difficult to calculate, interpret and compare among each other. An ICER cannot be interpreted without also knowing the quadrant of the cost-effectiveness plane in which the strategy lies, as ratio statistics will yield a positive ICER when there are: 1) cost savings and a reduction in QALYs, and 2) more costs but also QALYs gained. During analysis of multiple strategies, they are ranked by increasing effectiveness to calculate ICERs in reference to the less effective strategy. However, some strategies will need to be ruled out if they are extendedly dominated (i.e., the strategy has an ICER greater than a more effective alternative), and so the ICERs for the more effective alternative would need to be re-calculated each time once the extendedly dominated strategy is removed. The decision rule to identify the most cost-effective

strategy is unintuitive; it is not possible to rank strategies from most to least cost-effective using the ICER as the ratio statistics compares to one reference strategy at a time. In these situations, the NHB outcome can be used to present the cost-effectiveness of multiple strategies. The NHB approach does not use ratio statistics and has a natural unit measure of QALYs. This approach allows us to rank the strategies by their cost-effectiveness compared to each other, based on the highest number of QALYS (NHB) provided at a pre-specified cost-effectiveness threshold [12].

In this paper, we used the NHB approach and express all cost-effectiveness outcomes in NHB, which is expressed in units of QALYs. As such, we do not calculate or report ICERs in this study. The NHB framework has been commonly used to simplify cost-effectiveness results for decision-makers [13, 14]. The NHB of strategy n is defined as:

$$NHB_n = health_n - (cost_n/CET) \tag{1}$$

Where $health_n$ refers to the total amount of health resulting from strategy n (units: QALY), $cost_n$ refers to the total cost of strategy n (unit: $), and CET represents the cost-effectiveness threshold (unit: $/QALY). A positive NHB value represents a cost-effective intervention (i.e. effective trade-off between costs and health benefits for that strategy) at the chosen CET, and higher NHB values represent better value-for-money (i.e. more economically desirable strategies). In this analysis, we calculated NHB at commonly used CET of $50,000/QALY [15].

## Parameters and key assumptions

Table 1 outlines the base-case values and data sources for the parameter used in the model, which were obtained from published surveillance data and the literature.

**Diagnostic tests.**   The diagnostic test properties evaluated in this analysis were based on a recent meta-analysis [10]. We also considered a more recent meta-analysis [25], but did not incorporate its estimates because influenza A and B were not considered separately. All three types of rapid POC tests were associated with high specificities for adults (>96%) for influenza A and B. However, NAATs had the highest sensitivity for adults (87.4% for influenza A, 75.7% for influenza B), followed by DIAs (75.4% influenza A, 57% influenza B), and RIDTs (42.6% influenza A, 33.2% influenza B). Based on this meta-analysis, the sensitivity and specificity of Batch PCR were 100%, and results were available in 24 hours. However, for our base-case analysis, we assumed that Batch PCR sensitivity and specificity were slightly less than perfect at 95%. For the "Batch PCR–Wait" strategy, we assumed that the base-case probability of patients being treated within 48 hours of symptom onset (48%) [8], was reduced by half (24%) to account for the delayed results and potential start of treatment. For "Clinician Judgement", we used estimates from a study by Dugas and colleagues who assessed sensitivity (36%) and specificity (78%) in a high-risk population similar to our modeled population [16].

**Probabilities.**   Influenza epidemiology (e.g. prevalence, distribution of virus strains) was extracted from Canadian data sources for the base-case analysis. In the 2016–2017 surveillance season, the prevalence of influenza among patients presenting with ILI peaked at 14.4%. During this period, influenza A represented 87.3% of all laboratory-confirmed influenza cases [17]. We estimated hospitalization rates from an health administrative data study on influenza for influenza-confirmed patients in Canada [18]. We used hospitalization rates resulting from influenza only, and assumed that patients with ILI and not severely ill were not hospitalized (i.e., discharged home).

**Treatment.**   Given that oseltamivir represents the most commonly prescribed medication for influenza, we assumed that all patients received oseltamivir as NAI treatment. Patients infected with influenza who were treated with NAI in hospital had a reduced time to symptom

alleviation and lower risk of mortality relative to untreated patients [5, 8]. A recent meta-analysis of individual patient data was used to determine the proportion of patients treated within 48 hours of symptom onset (48%), and the reduction in mortality risk; the magnitude of reduction was stratified by the timing (early vs. late) of treatment [8]. The level of detail and generalizability offered by this meta-analysis made it suitable for use in the base-case analysis. Meta-analyses of randomized controlled trials have suggested no evidence of mortality benefit from oseltamivir[5, 26]. However, since these review includes RCTs of low-risk patients or mixed populations (i.e., the large majority of enrolled patients did not have severe influenza infection), the evidence is highly indirect for our study's target population. As such, we use the meta-analysis of observational studies, which is more direct to our study's target population, in the base-case and we conducted scenario analysis in which oseltamivir treatment has no mortality or hospitalization benefit. The probability of adverse events associated with treatment was estimated in a meta-analysis to be 7.5% (Nancy Santesso and colleagues, personal communication).

**QALYs and utilities.**    QALYs are calculated as the product of a utility and the number of life years gained. A utility is a numeric measure of the preference for a specific health state, and ranges between 0 (death) to 1 (perfect health), capturing the quality-of-life associated with the number of life years in a specific health state. We extracted QALY decrements for influenza from an economic evaluation in the United States, and assumed all patients with ILI or influenza received the same QALY decrement of 0.0146 to 0.0293 depending on their age (0.0293 for the base-case patient 65 years of age) [21, 27]. We assumed this decrement was constant over the episode of influenza, and that differential severity or length of stay would not significantly change the decrement. Benefit from oseltamivir treatment was estimated to be 0.00166 QALYs, based on time to symptom alleviation from the Cochrane review [5]. In the "Batch PCR–Treat", we assumed that one day of NAI treatment regimen prior to test results becoming available do not provide QALY benefit to patients testing negative. Accrued lifetime QALYs were calculated using utilities from a community-dwelling population between 0.88 and 0.94 [20].

**Costs.**    All direct costs were extracted from the literature and inflated to 2017 Canadian dollars. For diagnostic tests costs, we used the lower limit of range estimates from Merckx et al. for RIDT ($20), DIA ($20), and NAATs ($40) [10]. We assumed the hospital setting invested in start-up and capital costs for all diagnostic tests and rapid diagnostic tests did not require lab technician time. All healthcare utilization costs related to influenza were extracted from an administrative data study from the Canadian Immunisation Research Network [18]. A complete course of oseltamivir treatment was $42, which was calculated using Ontario Drug Benefit list prices and current recommended treatment algorithms [9, 24].

## Analysis

The base-case analysis was conducted for adult patients 65 years of age presenting to the ED with ILI, with a seasonal pre-test probability of influenza of 14.4% and a seasonal influenza A probability of 87.3%. The probability of being hospitalized was 11.6% and the probability of being treated with oseltamivir within 48 hours of symptom onset was 48%. We assumed all patients testing positive for influenza were given oseltamivir based on current treatment recommendations, and that adverse events did not extend length of stay or increase healthcare utilization.

We assessed cost-effectiveness in multiple scenario analyses: best-case (e.g. upper limit of test characteristics) and worst-case (e.g. lower limit) scenarios for all diagnostic tests, a scenario where cost of adverse events were equivalent to the cost of an ED visit, a scenario where

treatment with oseltamivir does not reduce risk of hospitalization or mortality, a scenario where 50% of individuals who test negative for influenza are still given treatment, and use of diagnostics in children (5 years of age) with the appropriate data. We conducted extensive deterministic sensitivity analysis to assess parameter uncertainty (e.g., pre-test probability for influenza since epidemiology varies seasonally and regionally). We assigned beta distributions for probabilities and utilities, and gamma distributions for costs to perform a probabilistic sensitivity analysis using 100,000 Monte Carlo simulations. We reported results following the Consolidated Health Economic Evaluation Reporting Standards (CHEERS) Guidelines (S1 File) [28].

**Sources of funding.** This work was partially supported by the World Health Organization. The funders had no role in study design, data collection and analysis, decision to publish, or preparation of the manuscript. S. M. is supported by a CIHR Frederick Banting and Charles Best Canada Graduate Scholarship Doctoral Award GSD-159274. B.S. is supported by a Canada Research Chair in Economics of Infectious Diseases (CRC-950-232429). There was no additional external funding received for this study.

## Results

### Base-case analysis

The preferred strategies by increasing NHB (lowest to highest) are: "Don't Treat Anyone", "Clinical Judgement", "RIDT", "Batch PCR–Wait", "DIA", "NAAT", "Batch PCR–Treat" and "Treat Everyone". "Treat Everyone" resulted in the highest number of expected QALYs per patient at 15.0477 at an expected cost of $630.01. At a CET of $50,000/QALY, "Treat Everyone" resulted in the highest NHB (15.0344 QALYs), and was considered the most cost-effective strategy in the high-risk older population. The NHB for "Batch PCR—Treat" and "NAAT" strategies were 15.0318, and 15.0277 QALYs, respectively. All results are summarized in Table 2. "Batch PCR–Treat" and "NAAT" resulted in reduced NHBs due to reduced health

**Table 2. Base-case results.**

| Strategy | Health Outcomes (Proportions) [†] | | | | Health Outcomes (per 100,000) | | | Cost-effectiveness | | |
|---|---|---|---|---|---|---|---|---|---|---|
| | Patients with influenza | | Patients without influenza | | Adverse events | Hospitalisations | Mortality | QALYs | Costs (CAD) | NHB at $50,000 CET (QALYs) |
| | Appropriate (Tx-Flu) | Inappropriate (No Tx-Flu) | Appropriate (No Tx-No Flu) | Inappropriate (Tx–No Flu) | | | | | | |
| "Don't Treat Anyone" | 0.00 | 1.00 | 1.00 | 0.00 | 404 | 1,680 | 1,836 | 14.9961 | 608.19 | 14.9839 |
| "Clinical Judgement" | 0.35 | 0.65 | 0.78 | 0.22 | 2,037 | 1,626 | 1,735 | 15.0145 | 611.02 | 15.0023 |
| "RIDT" | 0.41 | 0.59 | 1.00 | 0.00 | 712 | 1,619 | 1,715 | 15.0175 | 622.52 | 15.005 |
| "Batch PCR–Wait" | 0.95 | 0.05 | 0.95 | 0.05 | 1,362 | 1,595 | 1,659 | 15.0241 | 661.30 | 15.0109 |
| "DIA" | 0.73 | 0.27 | 0.97 | 0.03 | 1,113 | 1,569 | 1,604 | 15.0338 | 618.99 | 15.0214 |
| "NAAT" | 0.85 | 0.15 | 0.98 | 0.02 | 1,117 | 1,546 | 1,571 | 15.0404 | 636.75 | 15.0277 |
| "Batch PCR–Treat" | 0.95 | 0.05 | 0.00 | 1.00 | 2,348 | 1,522 | 1,537 | 15.0450 | 661.19 | 15.0318 |
| "Treat Everyone" | 1.00 | 0.00 | 0.00 | 1.00 | 7,447 | 1,518 | 1,533 | 15.0470 | 630.01 | 15.0344 |

[†] Calculations are described in S2 File.

CAD, Canadian dollars; CET, cost-effectiveness threshold; DIA, digital immunoassay tests; Hosp., Hospitalization; NAAT, nucleic acid amplification test; NHB, Net health benefit; PCR, polymerase chain reaction; QALYs, quality-adjusted life years; RIDT, rapid influenza diagnostic tests; Tx, treatment

benefits (e.g. increased hospitalization, mortality) compared to "Treat Everyone", and increased diagnostic costs. However, "Batch PCR–Treat" resulted in 5,099 fewer adverse events (nausea/vomiting) per 100,000 persons when compared to "Treat Everyone" since the latter strategy does not confirm influenza diagnosis. "Treat Everyone" appropriately treated 100% of patients with influenza but inappropriately treated 100% of patients with ILI only (i.e., no influenza). Appropriateness of treatment outcomes are summarized in Fig 2.

Although "Don't Treat Anyone" was the least costly strategy ($608.91 per patient), it resulted in the highest number of hospitalizations (1,680 per 100,000), and deaths (1,836 per 100,000) which contributed to this strategy having the lowest NHB of 14.9839 QALYs. This strategy also resulted in the highest proportion of untreated influenza cases.

Of the three rapid diagnostic tests, using NAAT to inform NAI treatment ("NAAT") was the most cost-effective. This strategy resulted in the greatest health benefit, NHB and lowest number of deaths (1,571 deaths per 100,000) compared to "DIA" (1,604 deaths per 100,000) and "RIDT" (1,715 deaths per 100,000). "Clinical Judgement" was the least preferred method of diagnosis in terms of NHB when compared to RIDTs, DIAs, NAATs, and Batch PCR. Costs and effectiveness of all strategies are plotted on a cost-effectiveness plane in Fig 3.

## Sensitivity analysis

At a pretest probability of 0%, "Don't Treat Anyone" was the most cost-effective strategy based on the highest number of NHBs at 15.2689 QALYs (S1 Table). As the pre-test probability for influenza increased to 1%, the most cost-effective strategy changed to "Treat Everyone" followed by "Batch PCR–Treat". This order of preferred strategies remained constant as the pretest probability increased to 100%. Our model results were robust to the following variables within the ranges listed in Table 1: QALY improvement from NAI, disutility of adverse events, probability of treatment within 48 hours of symptom onset, probability of death (ICU or non-

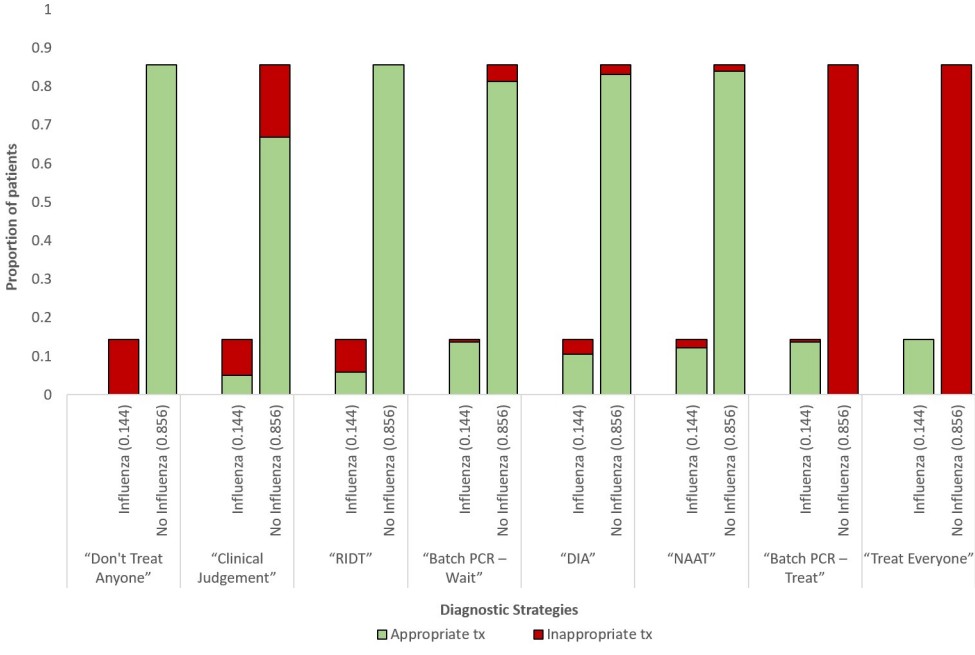

DIA, digital immunoassay; NAAT, nucleic acid amplification test; PCR, polymerase chain reaction; RIDT, rapid influenza diagnostic test; Tx, treatment.*Calculations are described in Appendix 2.

**Fig 2. Treatment appropriateness results for all strategies.**

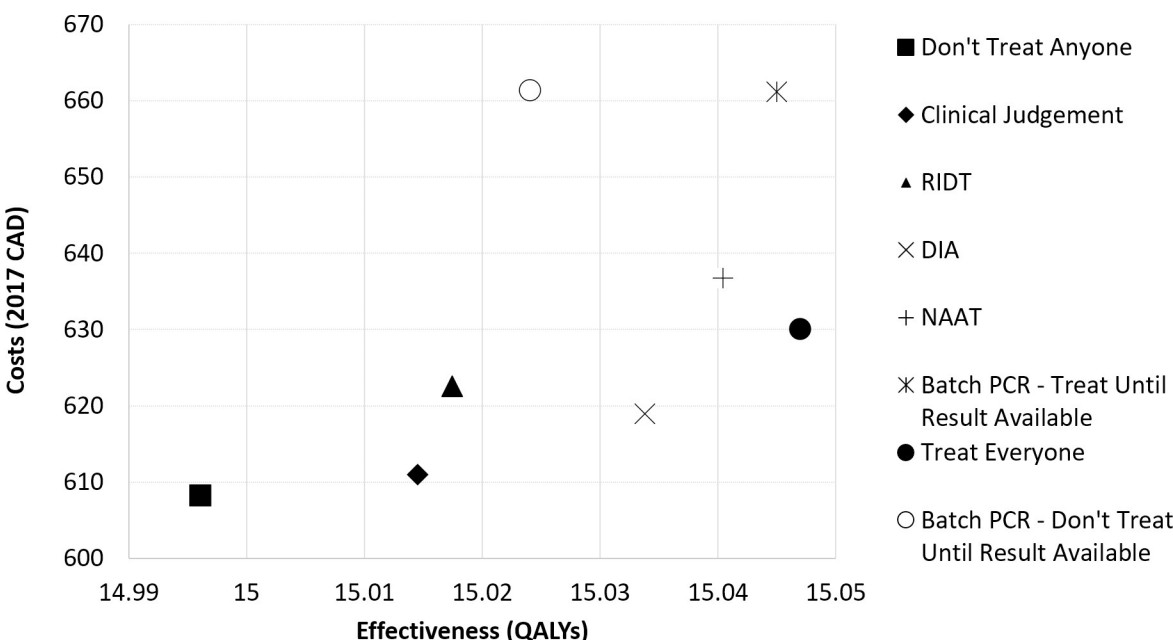

DIA, digital immunoassay; NAAT, nucleic acid amplification test; PCR, polymerase chain reaction; RIDT, rapid influenza diagnostic test

**Fig 3. Cost-effectiveness plane.**

ICU) after early treatment, probability of adverse events, cost of oseltamivir, and cost of batch PCR test.

Probabilistic sensitivity analysis determined that "Treat Everyone" strategy for the high-risk population was likely to be the most cost-effective strategy in over 85% of 100,000 simulations. A cost-effectiveness acceptability curve is included in S1 Fig.

## Scenario analysis

When the cost of AEs are assumed to be equivalent to the cost of an ED visit, the most cost-effective strategy remained "Treat Everyone", despite the average cost per patient increasing from $630 to $665, it afforded the highest NHB of 15.0337 QALYs. In children (corresponding data inputs in S2 Table), the preferred top three cost-effective strategies remain unchanged. In the scenario where early treatment (i.e., treatment within 48 hours of symptom onset) provided similar mortality benefits as late treatment (i.e., $\geq$ 48 hours post-symptom onset), "Treat Everyone" was still most cost-effective, followed by "Batch PCR–Treat". However, the NHB was considerably lower at 14.9916 QALYs compared to 15.0344 QALYs in the base-case for the most cost-effective strategy. In a subsequent scenario where oseltamivir treatment was assumed to provide no mortality, hospitalization or quality-of-life benefit, "Don't Treat Anyone" was the most cost-effective with a NHB of 14.9840 QALYs, followed by the "Clinical Judgement" with NHB of 14.9836 QALYs. In this scenario, "Treat Everyone" and "Batch PCR–Treat" have the lowest NHB.

When modeling diagnostic tests at their lowest sensitivity and specificity limits (i.e. worst case), the order of strategies' cost-effectiveness was unchanged from the base-case analysis. At the upper limits of sensitivity and specificity (i.e. best case), "Batch PCR–Treat" provided an incremental gain of 0.0014 QALYs over "Treat Everyone" at an incremental cost of $29.28. At a CET of $50,000/QALY, "Batch PCR–Treat" was equally as cost-effective as "Treat Everyone" with both strategies having a NHB of 15.0344 QALYs at a CET of $50,000/QALY. The cost-

effectiveness ranking of strategies utilizing diagnostic tests by NHB was: "Batch PCR–Treat", "NAAT", "DIA", "Batch PCR–Wait", "Clinical Judgement", and "RIDT". All scenario analysis results are summarized in S3 Table.

## Discussion

Based on our analysis, the preferred strategy in terms of health impact (QALYs) and cost-effectiveness (NHB) for high-risk older patients admitted to the ED presenting with ILI was "Treat Everyone", while the preferred diagnostic strategy to confirm influenza was "Batch PCR–Treat". While the "Treat Everyone" strategy is most cost-effective in terms of NHB, it precludes test results that are required to confirm influenza or to eliminate it as a diagnosis. "Batch PCR–Treat" was the second most cost-effective strategy. Similar to "Treat Everyone", this strategy starts high-risk severe patients on NAI therapy while awaiting test results, but continuation of treatment depends on the returned diagnostic result. This strategy reduces the number of individuals who are inappropriately treated and the number of adverse events. Of the three rapid diagnostic tests, "NAAT" resulted in the most optimal health outcomes (most QALYs, lowest number of deaths and inappropriate testing) and was cost-effective when compared to "DIA" and "RIDT". This was expected given that diagnostic test sensitivity was critical in identifying true influenza cases accurately for a quick start of antiviral NAI treatment benefit, and "NAAT" had the highest sensitivity of all rapid diagnostics at 0.87. While the NHB allowed us to determine the most cost-effective strategies, the differences in the average costs, QALYs, and NHB per patient between strategies are considered small in magnitude. For example, "Treat Everyone" resulted in a gain of 0.0066 QALYs compared to "NAAT", which is roughly equivalent to a gain of 2.4 days.

Sensitivity and scenario analysis suggested that while costs of treatment and diagnostics are important to consider in influenza management, they had little impact on the cost-effectiveness when compared to diagnostic test parameters, treatment benefits and seasonal prevalence of influenza. In a scenario analysis where the upper limit of sensitivity and specificity were used for all tests (i.e. the best-case scenario), "Batch PCR–Treat" was most preferred. These results suggested that the sensitivity and specificity are influential parameters to this model. A lower estimate of these test characteristics from the meta-analysis by Merckx and colleagues could have underestimated the strategies' cost-effectiveness. In this scenario, the sensitivity and specificity of Batch PCR for both influenza A and B were 1.00, which was the assumption used by Merckx and colleagues in their meta-analysis of rapid diagnostic tests [10].

In the literature, there have been several cost-effectiveness analyses of diagnostic testing for influenza, within the ED or hospital [23, 29–32]. Our results are comparable to a study by Dugas and colleagues in the United States, who assessed the cost-effectiveness of PCR-based rapid influenza testing and treatment using a decision-analytic model [29]. Dugas and colleagues concluded similar order of preferred strategies: treating all patients was most cost-effective and treating no patients with antivirals was the least. Dugas and colleagues used a QALY improvement of symptoms of 0.006, a pretest probability for influenza of 0.20 and evaluated high-risk patients who were 65 years of age. We used a similar but more conservative approach for QALY improvement due to NAI treatment (0.0017), pretest probability for influenza (0.15), and diagnostic sensitivity and specificity for batch PCR and clinical judgement.

In a Canadian study, Nshimyumukiza and colleagues estimated the cost-effectiveness of POC rapid tests versus clinical judgement in incremental costs per life-year saved for one seasonal influenza season, concluding that POC rapid tests were dominant compared to clinical judgement in Quebec, Canada [31]. We determined that "NAAT", "DIA" and "RIDT" were considered more cost-effective than "Clinical Judgement" based on NHB using a CET of

$50,000/QALY in Canada. Our study differs in that we report additional health outcomes, and cost-effectiveness using QALYs instead of life-years gained, and considered various testing strategies to guide treatment using updated diagnostic test characteristics.

Our analysis was subject to several limitations. These results should apply only to high-risk elderly patients presenting in the ED setting, and should not be extrapolated to lower risk populations or to other settings such as primary care, where risk of hospitalization, risk of mortality and cost of care may be lower. We did not incorporate resistance to antivirals or influenza transmission in our model, which are potential indirect consequences of the "Treat Everyone", and "Don't Treat Anyone" strategies, respectively. However, we estimated the proportion of high-risk older patients that would be appropriately and inappropriately treated with NAI in these strategies. While "Treat Everyone" resulted in higher NHB (i.e., cost-effectiveness), it inappropriately treats a large number of patients without influenza, which should be taken into consideration when comparing this strategy to "Batch PCR–Treat" during decision-making by both clinicians and policy-makers. As discussed previously, "Treat Everyone" may create antiviral resistance and lead to an unnecessary number of serious adverse events that may increase healthcare utilization. In addition, the incorrect use of the NAI therapy may be considered an opportunity cost where this volume of treatment could be used appropriately in other individuals presenting early with influenza. Clinicians value testing because it establishes a diagnosis and forces a focus on other possibilities if the test result is negative; this scenario was not modelled, which could understate the benefits of diagnostic testing prior to treating patients. However, severely ill patients with suspected influenza are typically treated with broad spectrum antibiotics and oseltamivir, pending results of multiple diagnostic tests. We assumed that healthcare facilities offering these diagnostic tests had negligible infrastructure and equipment costs, which may have underestimated the costs of newer diagnostics (e.g. NAAT and DIA).

Our model did not consider re-admissions or hospitalizations for other medical concerns. We assumed that high-risk older patients with ILI or influenza experience a similar disutility in quality-of-life (i.e., reduction in QALYs) and that influenza severity and hospitalization did not significantly alter the QALY decrement experienced by admitted patients. Since Muthuri and colleagues' meta-analysis on NAI treatment benefit on mortality was based on pandemic data, we conducted scenario analyses to examine the cost-effectiveness of strategies where early treatment did not confer additional mortality benefit than late treatment (which still provided some mortality benefit over no treatment). In this scenario, while the NHB between the strategies converged, the most cost-effective strategies remained the "Treat Everyone" followed by the "Batch PCR–Treat". In another scenario analysis where treatment with oseltamivir did not confer any mortality benefit, the "Don't Treat Anyone" strategy provided the most health outcomes and NHB (i.e., was most cost-effective) which is expected given treatment may result in adverse outcomes but provides no mortality, hospitalization or quality-of-life benefit. This finding along with the previous scenario where early treatment has no early benefit suggests that mortality benefit from treatment, regardless of early timing, drives the cost-effectiveness between the strategies.

Despite these limitations, this analysis comprehensively assessed the cost-effectiveness and impact of influenza point-of-care diagnostic tests on health outcomes in high-risk elderly patients admitted to the ED presenting with ILI. We reported results in QALYs, costs, and other health outcomes that are generalizable to other interventions and diagnostics for system level comparisons by decision-makers. Our analysis incorporated strong meta-analysis evidence on recently developed rapid diagnostic tests for influenza that have not been previously compared. Cost-effectiveness is an important consideration when implementing newer, more

costly, diagnostic technologies. These results are transferable in jurisdictions with similar influenza epidemiology, healthcare system (i.e. single payer system), and population health status.

## Conclusion

Treating high-risk older patients without performing a novel rapid diagnostic test resulted in the highest NHB and was the most cost-effective strategy. This strategy was less costly and reduced mortality through quicker and increased uptake of NAI. However, it inappropriately treats 100% of patients without influenza and does not provide diagnostic confirmation that can be attained by Batch PCR. Our analysis provides evidence on the impact of rapid diagnostic tests for influenza in the emergency department in terms of QALYs and cost-effectiveness that can be used by health policy decision-makers.

## Supporting information

**S1 File. CHEERS checklist.**
(PDF)

**S2 File. Calculations used for treatment appropriateness.**
(PDF)

**S1 Table. Sensitivity analyses results.**
(PDF)

**S2 Table. Parameters for children population.**
(PDF)

**S3 Table. Scenario analysis NHB results at a cost-effectiveness threshold of $50,000/ QALY.**
(PDF)

**S1 Fig. Cost-effectiveness acceptability curve.**
(PDF)

## Acknowledgments

We thank members of the WHO Guideline Development Group–Clinical Management of Severe Influenza Infections for comments on scenario development and interpretation made during a related guideline development meeting. The authors alone are responsible for the views expressed in this publication, and they do not necessarily represent the decisions, policies, or views of WHO.

## Author Contributions

**Conceptualization:** Stephen Mac, Ryan O'Reilly, Neill K. J. Adhikari, Robert Fowler, Beate Sander.

**Data curation:** Stephen Mac, Ryan O'Reilly, Neill K. J. Adhikari.

**Formal analysis:** Stephen Mac, Ryan O'Reilly, Robert Fowler, Beate Sander.

**Funding acquisition:** Neill K. J. Adhikari, Beate Sander.

**Methodology:** Stephen Mac, Ryan O'Reilly.

**Supervision:** Beate Sander.

**Writing – original draft:** Stephen Mac.

**Writing – review & editing:** Ryan O'Reilly, Neill K. J. Adhikari, Robert Fowler, Beate Sander.

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
