## [Decision Letter · Decision Letter 0]

5 Aug 2020

PONE-D-20-15378

Point-of-Care Diagnostic Tests for Influenza in the Emergency Department: A Cost-Effectiveness Analysis in a High-Risk Population from a Canadian Perspective

PLOS ONE

Dear Dr. Mac,

Thank you for submitting your manuscript to PLOS ONE. After careful consideration, we feel that it has merit but does not fully meet PLOS ONE’s publication criteria as it currently stands. Therefore, we invite you to submit a revised version of the manuscript that addresses the points raised during the review process.

We look forward to receiving your revised manuscript.

Kind regards,

Ruslan Kalendar, PhD

Academic Editor

PLOS ONE

Journal Requirements:

2. In your methods, please include a section describing the data used for this study including source, citation, description of the population represented by the data, and what categories of data were extracted.

3. In your ethics statement in the Methods section and in the online submission form, please provide additional information about the data used in your retrospective study. Specifically, please ensure that you have discussed whether all data were fully anonymized before you accessed them and/or whether the IRB or ethics committee waived the requirement for informed consent. If patients provided informed written consent to have data from their medical records used in research, please include this information.

4.Thank you for stating in your Funding Statement:

 [Sources of Funding: This work was partially supported by the World Health Organization.  The funders had no role in study design, data collection and analysis, decision to publish, or preparation of the manuscript.]. 

5.Thank you for stating the following in the Competing Interests section:

[I have read the journal's policy and the authors of this manuscript have the following competing interests: NKJA co-chaired the WHO Guideline Development Group – Clinical Management of Severe Influenza Infections.].

Reviewers' comments:

Reviewer's Responses to Questions

**Comments to the Author**

1. Is the manuscript technically sound, and do the data support the conclusions?

Reviewer #1: No

Reviewer #2: Yes

2. Has the statistical analysis been performed appropriately and rigorously? 

Reviewer #1: Yes

Reviewer #2: Yes

3. Have the authors made all data underlying the findings in their manuscript fully available?

Reviewer #1: Yes

Reviewer #2: Yes

4. Is the manuscript presented in an intelligible fashion and written in standard English?

Reviewer #1: Yes

Reviewer #2: Yes

5. Review Comments to the Author

Reviewer #1: General comments

There are two important problems with this manuscript. First, they base their assumption of mortality benefit on a very optimistic assessment from an observational study, and not on the best evidence from RCTs. Second, while their analysis is focused on a 65 year old presenting to the ED, that is not clearly emphasized throughout the abstract and discussion, giving readers the impression that it could apply to all patients in all settings.

Specific comments

Abstract

What is the $/QALY gained? The results are presented in such a way that the casual reader could infer that treating everyone results in 15 QALYs for a cost of only $600.

You state: “most favourable outcomes per cohort (1,571 deaths per 100,000).” Of course, death is not a favourable outcome, reducing deaths would be.

It should be clear that this analysi only applies to high risk adults presenting to the ED, and not to average risk adults presenting in primary care which is of course a far more common scenario. It is important that these results not be extrapolated, something the abstract as currently written wrould encourage as it says nothing about ED or high risk.

Introduction

Lines 33-35: But, no RCT evidence of mortality benefit (see Cochrane review by Jefferson and Ebell MH, Williamson M, Schofill, J. Effectiveness of oseltamivir in adults: a meta-analysis of published and unpublished clinical trials. Fam Pract 2013; Apr;30(2):125-33.

Line 40: Would add “with a high degree of accuracy” to this sentence. We have long had rapid tests, they just lacked sensitivity.

Line 43: “In recent years” would only apply to the point of care molecular tests, not RIDT.

Line 53-54

Methods

Lines 82-82: In the real world, unfortunately, many patients with negative tests still get a NAI.

Lines 85-86: Which meta-analyses? They should be cited. The two cited above found no difference in mortality based on RCT data. The one study (never published) by Roche of patients 65 and older found absolutely no symptom benefit. Your model hinges on a faulty assumption.

Lines 96-105: Please provide an example of how to ingerpret this. It is quite opaque to me, I’m used to seeing $/QALY or even $/quality adjusted life day for short-term events. No idea how to interpret NHB.

Table 1:

The sensitivities cited here are lower than those in the introduction (53% for RIDT) and also lower than those in other systematic reviews (~60%). This would bias against a testing strategy.

The probability of influenza of 0.144 is based, apparently, on seasonal prevalence in the community. Instead, it should be based on the prevalence of older patients presenting with ILI who have influenza. I suspect that is higher than 14%

Your mortality benefit is based on the IPD meta-analysis by Muthuri (Lancet Respir Med

. 2014 May;2(5):395-404). This was an observational study, not a SR of randomized trials. Of course it provides an optimistic estimate of benefit. The best available evidence from systematic reviews of RCTs (above) does not support a significant mortality benefit and at best only a small reduction in pneumonia.

Lines 92-93: This statement to the reader who is not a health economist (like me) makes it sound like there is an outlandishly large benefit at a very low cost (15 more QALY for $630). I know that isn’t true, but it is hard for the general reader to understand what is meant: “yielding an expected 15.0477 QALYs per patient, 193 at an expected cost of $630.01, and the greatest NHB of 15.0344 QALYs”. We are more accustomed to seeing ICERs.

Table 2.

Here I can see the ICER is about 0.05 additional QALYs at an additional cost of $22 for the treat everyone vs don’t treat anyone strategies.

Page 13, Discussion, para 1: You fail to say “high risk older patients” here.

Page 16, first para: “we conducted scenario analysis where the probability of death due to early treatment was equivalent to late treatment, showing that the order of test and treat strategies did not substantially change based on estimates of NAI effectiveness.” So, are you saying you modeled no mortality benefit and no reduction in hospitalizations, and it didn’t affect your results?

Page 16, 2nd para: In ‘high risk elderly patients”, not all patients coming to ED. You are over-generalizing your results. Obviously the high hospitalization risk of 12% means this ia very very different population from the general population with ILI. In the oseltamivir trials, the hospitalization rate was only 1.4%.

There is no funding declaration.

Reviewer #2:

 This paper describes excellent comparison of cost effectiveness of different treatment strategies used in cases of influenza like illness presented to the emergency department. The authors have very well compared these strategies and discussed well the impact of the top most strategies. This provides an useful information on cost efficacy of different treatment approaches in such situation.

In discussion, the side effects of approach "Treat everyone" should be discussed in more detail with other published findings. In my view, that is an important piece to discuss the side effects of treatment that could happen in patients without the illness if that approach is recommended.

The comparison of "Batch PCR-Treat" and "Treat Everyone" could be highlighted more.

6. PLOS authors have the option to publish the peer review history of their article (what does this mean?). If published, this will include your full peer review and any attached files.

Reviewer #1: **Yes: **Mark H. Ebell MD, MS, Professor of Epidemiology, University of Georgia, Athens, USA

Reviewer #2: No

---

## [Author Response · Author response to Decision Letter 0]

22 Sep 2020

Stephen Mac

Institute of Health Policy, Management, and Evaluation

University of Toronto

155 College St. Suite 425

Toronto, ON M5T3M6

Email: sm.mac@mail.utoronto.ca

To: Ruslan Kalendar

Academic Editor

PLOS ONE

September 16, 2020

RE: PONE-D-20-15378 “Point-of-Care Diagnostic Tests for Influenza in the Emergency Department: A Cost-Effectiveness Analysis in a High-Risk Population from a Canadian Perspective”

Dear Dr. Kalendar and Reviewers,

Thank you for taking the time to review our manuscript and providing insightful comments and suggestions. We have taken into consideration all comments and suggestions highlighted by the reviewers and revised our manuscript accordingly for resubmission to PLOS One. Please find enclosed our revised manuscript entitled “Point-of-Care Diagnostic Tests for Influenza in the Emergency Department: A Cost-Effectiveness Analysis in a High-Risk Population from a Canadian Perspective”. 

Below we summarize the key issues that were raised, our explanations (if applicable), how we addressed them, and where the revisions were made in the revised manuscript with tracked changes.

1. Treatment effect. There was concern from Reviewer 1 that our model’s use of the meta-analysis from Muthuri et al. was faulty since other reviews of randomized clinical trials have shown no mortality and hospitalization benefits. 

Explanation: We decided to use the meta-analysis of observational studies (Muthuri et al.) since the reviews of randomised controlled trials (RCTs) do not only include high-risk patients (i.e., the large majority of enrolled patients did not have severe influenza infection), and therefore the evidence from them is highly indirect for our study’s target population (high-risk older population presenting to the ED); so that the direct evidence from observational studies is more appropriate to use, even though it is downgraded for risk of bias using the appropriate GRADE assessment for study design. This was the position taken at the time of study conceptualization by the Guideline Development Group (GDG) members for the World Health Organization (WHO) guideline. We do acknowledge that this is a debatable point but believe that our rationale is sufficient to support the use of the meta-analysis of observational studies for mortality benefit in our study.

Revisions: We have added this rationale and acknowledged that there are high-quality studies with a mixed population, that suggest no mortality benefit, into the Methods section (p.10-11, Ln 176-182) of the revised manuscript. Furthermore, we have simulated a scenario analysis, which is outlined in the Methods of the revised manuscript (p.12, Ln 219-221), where we assume that the NAI has no mortality, hospitalization benefit (i.e., citing the RCT mentioned in the comments) or quality-of-life benefit. The results suggest that “Treat Everyone” is not cost-effective and that “Don’t Treat Anyone” (i.e., not testing and not treating) is the most cost-effective in this scenario. We have included the results of this additional analysis in the Results (p.16, Ln 292-296), summarized in S6 Table, and elaborated on the implications of these findings in the Discussion (p.20, Ln 381-387) of the revised manuscript.

2. The use of net health benefit (NHB) instead of ICERs. There was confusion as to what the NHB actually represents in this paper and the omission of ICERs, which the reviewer is most familiar with cost-effectiveness analyses.

Explanation: We understand that typically cost-effectiveness analyses (CEA) report the value of interventions or technologies in terms of incremental cost-effectiveness ratios (ICER) but in our study where we compare eight strategies, we decided to report cost-effectiveness using the net health benefit approach (NHB). 

In CEA, there are various units of measure to present cost-effectiveness: using incremental cost-effectiveness ratios (ICER) taking a cost-utility approach which would express value in a $/QALY gained, net monetary benefit (NMB) which would express value in terms of costs, and net health benefits (NHB) which expresses value in terms of the health outcome chosen (i.e., in quality-adjusted life years in this study). (Paulden, M., PharmacoEconomics (2020) 38:781–784).

While most conventional CEA in the literature report value in terms of ICERs of cost per QALY gained, it is typically preferred when comparing 2-3 strategies. As the number of strategies being compared increases, the ratio statistics of the ICER become more difficult to calculate, interpret and compare amongst each other. An ICER cannot be interpreted without also knowing the quadrant of the cost-effectiveness plane in which the strategy lies, as ratio statistics will yield a positive ICER when there are: 1) cost savings and a reduction in QALYs, and 2) more costs but also QALYs gained. Some strategies will need to be ruled out if they are extendedly dominated, and the ICERs would need to be re-calculated depending on the reference strategy. The decision rule to identify the most cost-effective strategy is unintuitive; it is not possible to rank strategies from most to least cost-effective using the ICER as the ratio statistics compares to one reference strategy at a time. (Paulden, M., PharmacoEconomics (2020) 38:781–784).

In these situations, the NHB outcome can be used to present the cost-effectiveness of multiple strategies. The NHB approach does not use ratio statistics and has a natural unit measure of QALYs. This approach allows us to rank the strategies by their cost-effectiveness compared to each other, based on the highest number of QALYS (NHB) provided at a pre-specified cost-effectiveness threshold.

Revisions: We recognize that it may be confusing throughout the study by reporting the costs and then the NHB, which has a natural unit of QALYs. Therefore, we have taken the necessary steps below to ensure the reader understands the use of the NHB approach. Throughout the revised manuscript, we have:

• Removed any mention of the ICER (i.e., we do not report the $/QALY gained anywhere as this approach was not used). 

• Reported the strategies’ outcomes more consistently and distinctly in terms of costs, QALYs, and NHB 

• Further elaborated on the NHB approach in the Methods section (p.6-7, Ln103-135) and the rationale for using this as opposed to the ICER approach. 

Following this letter, we included a table outlining our detailed response to all comments for your review. Thank you for your time and consideration. We look forward to your decision. 

Sincerely,

Stephen Mac 

On behalf of all authors below

 

Stephen Mac PhD(c), MBiotech

Institute of Health Policy, Management and Evaluation, University of Toronto, Toronto, Canada

Toronto Health Economics and Technology Assessment (THETA) Collaborative, University Health Network, Toronto, Canada

Ryan O’Reilly MD(c), PhD(c)

Department of Medicine, McMaster University, Hamilton, Canada

Institute of Health Policy, Management and Evaluation, University of Toronto, Toronto, Canada

Toronto Health Economics and Technology Assessment (THETA) Collaborative, University Health Network, Toronto, Canada

Neill Adhikari MDCM, MSc

Department of Critical Care Medicine, Sunnybrook Health Sciences Centre, Toronto, Canada

Interdepartmental Division of Critical Care Medicine, University of Toronto, Toronto, Canada

Rob Fowler MDCM, FRCPC, MSc

Institute of Health Policy, Management and Evaluation, University of Toronto, Toronto, Canada

Department of Critical Care Medicine, Sunnybrook Health Sciences Centre, Toronto, Canada

Interdepartmental Division of Critical Care Medicine, University of Toronto, Toronto, Canada

Beate Sander PhD

Institute of Health Policy, Management and Evaluation, University of Toronto, Toronto, Canada

Toronto Health Economics and Technology Assessment (THETA) Collaborative, University Health Network, Toronto, Canada

Public Health Ontario, Toronto, Canada

ICES, Toronto, Canada

 

Reviewer 1 

Abstract 

What is the $/QALY gained? The results are presented in such a way that the casual reader could infer that treating everyone results in 15 QALYs for a cost of only $600.

Response: We understand that typically cost-effectiveness analyses (CEA) report the value of interventions or technologies in terms of incremental cost-effectiveness ratios (ICER) but in our study where we compare eight strategies, we decided to report cost-effectiveness using the net health benefit approach (NHB). 

In CEA, there are various units of measure to present cost-effectiveness: using incremental cost-effectiveness ratios (ICER) taking a cost-utility approach which would express value in a $/QALY gained, net monetary benefit (NMB) which would express value in terms of costs, and net health benefits (NHB) which expresses value in terms of the health outcome chosen (i.e., in quality-adjusted life years in this study). (Paulden, M., PharmacoEconomics (2020) 38:781–784).

While most conventional CEA in the literature report value in terms of ICERs of cost per QALY gained, it is typically preferred when comparing 2-3 strategies. As the number of strategies being compared increases, the ratio statistics of the ICER become more difficult to calculate, interpret and compare amongst each other. An ICER cannot be interpreted without also knowing the quadrant of the cost-effectiveness plane in which the strategy lies, as ratio statistics will yield a positive ICER when there are: 1) cost savings and a reduction in QALYs, and 2) more costs but also QALYs gained. Some strategies will need to be ruled out if they are extendedly dominated, and the ICERs would need to be re-calculated depending on the reference strategy. The decision rule to identify the most cost-effective strategy is unintuitive; it is not possible to rank strategies from most to least cost-effective using the ICER as the ratio statistics compares to one reference strategy at a time. (Paulden, M., PharmacoEconomics (2020) 38:781–784).

In these situations, the NHB outcome can be used to present the cost-effectiveness of multiple strategies. The NHB approach does not use ratio statistics and has a natural unit measure of QALYs. This approach allows us to rank the strategies by their cost-effectiveness compared to each other, based on the highest number of QALYS (NHB) provided at a pre-specified cost-effectiveness threshold.

We recognize that it may be confusing throughout the study by reporting the costs and then the NHB, which has a natural unit of QALYs. Therefore, we have taken the necessary steps below to ensure the reader understands the use of the NHB approach. Throughout the revised manuscript, we have:

• Removed any mention of the ICER (i.e., we do not report the $/QALY gained anywhere as this approach was not used). 

• Reported the strategies’ outcomes more consistently and distinctly in terms of costs, QALYs, and NHB 

• Further elaborated on the NHB approach in the Methods section (p.6-7, Ln103-135) and the rationale for using this as opposed to the ICER approach.

You state: “most favourable outcomes per cohort (1,571 deaths per 100,000).” Of course, death is not a favourable outcome, reducing deaths would be. 

Response: We have re-worded this to read: “…the least number of deaths (1,571 per 100,000)” in the revised manuscript (p. 1, Ln 16-17).

It should be clear that this analysis only applies to high risk adults presenting to the ED, and not to average risk adults presenting in primary care which is of course a far more common scenario. It is important that these results not be extrapolated, something the abstract as currently written would encourage as it says nothing about ED or high risk. 

Response: Thank you for pointing this out. We have added the target population (high-risk adults) into the Abstract in the revised manuscript (p.1, Ln 8, 20).

Introduction 

Lines 33-35: But no RCT evidence of mortality benefit (see Cochrane review by Jefferson and Ebell MH, Williamson M, Schofill, J. Effectiveness of oseltamivir in adults: a meta-analysis of published and unpublished clinical trials. Fam Pract 2013; Apr;30(2):125-33.

Response: Thank you for this comment and let us explain our rationale. We decided to use the meta-analysis of observational studies (Muthuri et al.) since the reviews of randomised controlled trials (RCTs) do not only include high-risk patients (i.e., the large majority of enrolled patients did not have severe influenza infection), and therefore the evidence from them is highly indirect for our study’s target population (high-risk older population presenting to the ED); so that the direct evidence from observational studies is more appropriate to use, even though it is downgraded for risk of bias using the appropriate GRADE assessment for study design. This was the position taken at the time of study conceptualization by the Guideline Development Group (GDG) members for the World Health Organization (WHO) guideline. We do acknowledge that this is a debatable point but believe that our rationale is sufficient to support the use of the meta-analysis of observational studies for mortality benefit in our study.

We have added this rationale and acknowledged that there are high-quality studies with a mixed population, that suggest no mortality benefit, into the Methods section (p.10-11, Ln 176-182) of the revised manuscript. Furthermore, we have simulated a scenario analysis, which is outlined in the Methods of the revised manuscript (p.12, Ln 219-221), where we assume that the NAI has no mortality, hospitalization benefit (i.e., citing the RCT mentioned in the comments) or quality-of-life benefit. The results suggest that “Treat Everyone” is not cost-effective and that “Don’t Treat Anyone” (i.e., not testing and not treating) is the most cost-effective in this scenario. We have included the results of this additional analysis in the Results (p.16, Ln 292-296), summarized in S6 Table, and elaborated on the implications of these findings in the Discussion (p.20, Ln 381-387) of the revised manuscript.

Line 40: Would add “with a high degree of accuracy” to this sentence. We have long had rapid tests, they just lacked sensitivity. Added as suggested in the revised manuscript (p.2, Ln 41).

Line 43: “In recent years” would only apply to the point of care molecular tests, not RIDT. 

Response: Removed words as suggested in the revised manuscript (p.2, Ln 46).

Methods 

Lines 82-82: In the real world, unfortunately, many patients with negative tests still get a NAI.

Response: We acknowledge that possibility and have assumed for the base-case results that these patients with negative tests do not get a NAI. We have revised in the Methods (p.5, Ln 87-90; p.12, Ln220-221). We also conducted scenario analysis where 50% of patients with a negative test still received a NAI. Results are summarized in S6 Table. 

Lines 85-86: Which meta-analyses? They should be cited. The two cited above found no difference in mortality based on RCT data. The one study (never published) by Roche of patients 65 and older found absolutely no symptom benefit. Your model hinges on a faulty assumption. 

Response: We have cited the meta-analyses used in the revised manuscript (p.5, Ln 93). Regarding our model assumption, please see the comments above for our rationale to use this meta-analysis for mortality benefit, and the other for calculation of QALY benefit. We do acknowledge that this is a debatable point but believe that our rationale is sufficient to support the use of the meta-analysis of observational studies for mortality benefit in our study.

Lines 96-105: Please provide an example of how to interpret this. It is quite opaque to me; I’m used to seeing $/QALY or even $/quality adjusted life day for short-term events. No idea how to interpret NHB.

Response: As discussed above, we have further elaborated on the rationale of using the NHB approach, how to calculate the outcomes (QALYs), and interpret it as a cost-effectiveness measure in the Methods section. We hope that this will help readers understand how to use and interpret the NHB to understand the cost-effectiveness of each of the strategies. 

Table 1: The sensitivities cited here are lower than those in the introduction (53% for RIDT) and also lower than those in other systematic reviews (~60%). This would bias against a testing strategy.

Response: Thank you for pointing this out. We based all sensitivity and specificity parameters from the Merckx et al. systematic review and meta-analysis. In Table 1, the test parameters are specifically for adults whereas the parameters in the Introduction are in total over all populations. We have revised in Table 1 (p.8-9) to include “adult”, and the Introduction (p.4, Ln 53) to clarify that these parameters are for the entire meta-analysis population: “adults and children”. We acknowledge that there are other systematic reviews and studies that suggest higher or lower test parameters. However, the Merckx et al study was considered a high-quality study. We conducted a scenario analysis to address the uncertainty of test parameters in the Results (p.16-17, Ln 298-306), elaborated further on the importance of sensitivity and specificity in the Discussion (p.17, Ln 327-330), and summarized all results in S6 Table. 

The probability of influenza of 0.144 is based, apparently, on seasonal prevalence in the community. Instead, it should be based on the prevalence of older patients presenting with ILI who have influenza. I suspect that is higher than 14%

Response: We agree that the probability of influenza is a seasonal prevalence estimate in the community that can fluctuate from season-to-season and depending on the population of interest. To address this, we conducted sensitivity analysis to explore the impact of this uncertainty on the study results. This is reported in the Results (p.15, Ln 270-274), and S3 Table.

Your mortality benefit is based on the IPD meta-analysis by Muthuri (Lancet Respir Med. 2014 May;2(5):395-404). This was an observational study, not a SR of randomized trials. Of course, it provides an optimistic estimate of benefit. The best available evidence from systematic reviews of RCTs (above) does not support a significant mortality benefit and at best only a small reduction in pneumonia.

Response: We do acknowledge that this is a debatable point but believe that our rationale is sufficient to support the use of the meta-analysis of observational studies for mortality benefit in our study. Please see comments above. 

Lines 92-93: This statement to the reader who is not a health economist (like me) makes it sound like there is an outlandishly large benefit at a very low cost (15 more QALY for $630). I know that isn’t true, but it is hard for the general reader to understand what is meant: “yielding an expected 15.0477 QALYs per patient, 193 at an expected cost of $630.01, and the greatest NHB of 15.0344 QALYs”. We are more accustomed to seeing ICERs. 

Response: Thank you for pointing this out. As discussed above, we have revised the manuscript throughout to describe all strategies’ outcomes more consistently and distinctly in terms of costs, QALYs, and NHB to avoid confusion for audience who are not health economists. We will only report NHB outcomes in this study to avoid going back and forth and confusion to the readers.

Table 2. Here I can see the ICER is about 0.05 additional QALYs at an additional cost of $22 for the treat everyone vs doesn’t treat anyone strategies. 

Response: As discussed above, we will only report NHB outcomes in this study to avoid going back and forth and confusing the readers. 

Discussion 

Page 13, Discussion, para 1: You fail to say “high risk older patients” here. 

Response: Added as suggested in the revised manuscript (p.17, Ln 310).

Page 16, first para: “we conducted scenario analysis where the probability of death due to early treatment was equivalent to late treatment, showing that the order of test and treat strategies did not substantially change based on estimates of NAI effectiveness.” So, are you saying you modeled no mortality benefit and no reduction in hospitalizations, and it didn’t affect your results?

Response: Thank you for this comment. This sentence as it is currently written can be misinterpreted. In this scenario, we assume that the mortality benefit of early treatment is equal to the mortality benefit of late treatment (i.e., early treatment does not provide additional mortality benefit if within 48 hours of symptom onset). However, there is still some mortality benefit with late treatment vs. no treatment. While the ordering of the strategies in term of cost-effectiveness did not change, the NHB of each strategy was reduced. We have made these results and discussion/explanation clearer in the Discussion (p.20, Ln 376-387).

Page 16, 2nd para: In ‘high risk elderly patients”, not all patients coming to ED. You are over-generalizing your results. Obviously, the high hospitalization risk of 12% means this is a very different population from the general population with ILI. In the oseltamivir trials, the hospitalization rate was only 1.4%. 

Response: Thank you for pointing this out. We have added the target population (high-risk elderly patients) so that we are not overgeneralizing, and readers understand that this is for a specific population with higher hospitalization rates. The changes are in the revised manuscript (p.20, Ln 391).

There is no funding declaration. 

Response: This was stated at the end of the Abstract. We have now added the funding declaration into the Methods as suggested in the revised manuscript (p.13, Ln 229-231).

 

Reviewer 2 

In discussion, the side effects of approach "Treat everyone" should be discussed in more detail with other published findings. In my view, that is an important piece to discuss the side effects of treatment that could happen in patients without the illness if that approach is recommended.

Response: We agree with this comment and have tried to outline some of the possible adverse events and consequences of the “Treat Everyone” strategy in the discussion. We have further elaborated on this in the revised manuscript (p.19, Ln 359-364).

The comparison of "Batch PCR-Treat" and "Treat Everyone" could be highlighted more. 

Response: Thank you for this comment. We have further compared the “Batch PCR – Treat” and “Treat Everyone” strategies in the discussion of the revised manuscript (p.17, Ln 311-317).

---

## [Decision Letter · Decision Letter 1]

26 Oct 2020

PONE-D-20-15378R1

Point-of-care diagnostic tests for influenza in the emergency department: A cost-effectiveness analysis in a high-risk population from a Canadian perspective

PLOS ONE

Dear Dr. Mac,

Thank you for submitting your manuscript to PLOS ONE. After careful consideration, we feel that it has merit but does not fully meet PLOS ONE’s publication criteria as it currently stands. Therefore, we invite you to submit a revised version of the manuscript that addresses the points raised during the review process.

Authors need to prepare a manuscript in accordance with the reviewer's comment.

We look forward to receiving your revised manuscript.

Kind regards,

Ruslan Kalendar, PhD

Academic Editor

PLOS ONE

Reviewers' comments:

Reviewer's Responses to Questions

**Comments to the Author**

Reviewer #1: (No Response)

2. Is the manuscript technically sound, and do the data support the conclusions?

Reviewer #1: Yes

3. Has the statistical analysis been performed appropriately and rigorously? 

Reviewer #1: Yes

4. Have the authors made all data underlying the findings in their manuscript fully available?

Reviewer #1: Yes

5. Is the manuscript presented in an intelligible fashion and written in standard English?

Reviewer #1: Yes

6. Review Comments to the Author

Reviewer #1: The authors have done a good job of responding to comments. I have a few final recommendations:

1. The authors write:

Other meta-analysis in the literature have suggested no evidence of mortality benefit from oseltamivir[26]. However, since this review includes RCTs of low-risk patients or mixed populations (i.e., the large majority of enrolled patients did not have severe influenza infection), the evidence is highly indirect for our study’s target population.

I suggest revising this as follows and adding reference to the Cochrane Review by Jefferson and colleagues.

Meta-analyses of randomized controlled trials have suggested no evidence of mortality benefit from oseltamivir[26,<insert cochrane="" reference="">]. However, since this review includes RCTs of low-risk patients or mixed populations (i.e., the large majority of enrolled patients did not have severe influenza infection), the evidence is highly indirect for our study’s target population.

2. There should be a clear statement to the effect that these results apply only to high risk elderly patients being evaluated in the ED setting and should not be extrapolated to lower risk populations or to other settings such as primary care, where risk of mortality and hospitalization and cost of care are all much lower, but the adverse events related to the drug and cost of the drug remain the same.

3. There should also be a discussion of the magnitude of the difference in QALYs, costs, and net health benefit between strategies. For example, in Table 2 the QALYs for NAAT strategy were 15.0404 and for treat everyone were 15.0470. This is a gain of only 0.0066 QALYs or 2.4 days on average, which is quite small.</insert>

---

## [Author Response · Author response to Decision Letter 1]

28 Oct 2020

Stephen Mac

Institute of Health Policy, Management, and Evaluation

University of Toronto

155 College St. Suite 425

Toronto, ON M5T3M6

Email: sm.mac@mail.utoronto.ca

To: Ruslan Kalendar

Academic Editor

PLOS ONE

October 28, 2020

RE: PONE-D-20-15378-R1 “Point-of-Care Diagnostic Tests for Influenza in the Emergency Department: A Cost-Effectiveness Analysis in a High-Risk Population from a Canadian Perspective”

Dear Dr. Kalendar,

Thank you for taking the time to review our manuscript and providing additional comments. We have addressed all comments and suggestions and revised our manuscript accordingly for resubmission to PLOS One. Please find enclosed our revised manuscript entitled “Point-of-Care Diagnostic Tests for Influenza in the Emergency Department: A Cost-Effectiveness Analysis in a High-Risk Population from a Canadian Perspective”. 

Following this letter, we included a table outlining our detailed response to all comments for your review. Thank you for your time and consideration. We look forward to your final decision. 

Sincerely,

Stephen Mac 

On behalf of all authors below

 

Stephen Mac PhD(c), MBiotech

Institute of Health Policy, Management and Evaluation, University of Toronto, Toronto, Canada

Toronto Health Economics and Technology Assessment (THETA) Collaborative, University Health Network, Toronto, Canada

Ryan O’Reilly MD(c), PhD(c)

Department of Medicine, McMaster University, Hamilton, Canada

Institute of Health Policy, Management and Evaluation, University of Toronto, Toronto, Canada

Toronto Health Economics and Technology Assessment (THETA) Collaborative, University Health Network, Toronto, Canada

Neill Adhikari MDCM, MSc

Department of Critical Care Medicine, Sunnybrook Health Sciences Centre, Toronto, Canada

Interdepartmental Division of Critical Care Medicine, University of Toronto, Toronto, Canada

Rob Fowler MDCM, FRCPC, MSc

Institute of Health Policy, Management and Evaluation, University of Toronto, Toronto, Canada

Department of Critical Care Medicine, Sunnybrook Health Sciences Centre, Toronto, Canada

Interdepartmental Division of Critical Care Medicine, University of Toronto, Toronto, Canada

Beate Sander PhD

Institute of Health Policy, Management and Evaluation, University of Toronto, Toronto, Canada

Toronto Health Economics and Technology Assessment (THETA) Collaborative, University Health Network, Toronto, Canada

Public Health Ontario, Toronto, Canada

ICES, Toronto, Canada

 

Reviewer [Response]

The authors write: Other meta-analysis in the literature have suggested no evidence of mortality benefit from oseltamivir[26]. However, since this review includes RCTs of low-risk patients or mixed populations (i.e., the large majority of enrolled patients did not have severe influenza infection), the evidence is highly indirect for our study’s target population.

I suggest revising this as follows and adding reference to the Cochrane Review by Jefferson and colleagues.

Meta-analyses of randomized controlled trials have suggested no evidence of mortality benefit from oseltamivir[26,]. However, since this review includes RCTs of low-risk patients or mixed populations (i.e., the large majority of enrolled patients did not have severe influenza infection), the evidence is highly indirect for our study’s target population.

RESPONSE: [We have revised as suggested on page 9 (line 170 to 172).]

There should be a clear statement to the effect that these results apply only to high risk elderly patients being evaluated in the ED setting and should not be extrapolated to lower risk populations or to other settings such as primary care, where risk of mortality and hospitalization and cost of care are all much lower, but the adverse events related to the drug and cost of the drug remain the same.

RESPONSE: [We have revised as suggested in the limitations of the Discussion on page 19 (line 346 to 349).]

There should also be a discussion of the magnitude of the difference in QALYs, costs, and net health benefit between strategies. For example, in Table 2 the QALYs for NAAT strategy were 15.0404 and for treat everyone were 15.0470. This is a gain of only 0.0066 QALYs or 2.4 days on average, which is quite small.

RESPONSE: [We have revised as suggested in the first paragraph of the Discussion on page 17 (line 311 to 314).]

---

## [Editor Report · Decision Letter 2]

30 Oct 2020

Point-of-care diagnostic tests for influenza in the emergency department: A cost-effectiveness analysis in a high-risk population from a Canadian perspective

PONE-D-20-15378R2

Dear Dr. Mac,

We’re pleased to inform you that your manuscript has been judged scientifically suitable for publication and will be formally accepted for publication once it meets all outstanding technical requirements.

Kind regards,

Ruslan Kalendar, PhD

Academic Editor

PLOS ONE

---

## [Editor Report · Acceptance letter]

6 Nov 2020

PONE-D-20-15378R2 

Point-of-care diagnostic tests for influenza in the emergency department: A cost-effectiveness analysis in a high-risk population from a Canadian perspective 

Dear Dr. Mac:

I'm pleased to inform you that your manuscript has been deemed suitable for publication in PLOS ONE. Congratulations! Your manuscript is now with our production department. 

Kind regards, 

on behalf of

Dr. Ruslan Kalendar 

Academic Editor

PLOS ONE